# Prior Exposure and Toddlers’ Sleep-Related Memory for Novel Words

**DOI:** 10.3390/brainsci11101366

**Published:** 2021-10-18

**Authors:** Emma L. Axelsson, Jaclyn Swinton, Isabel Y. Jiang, Emma V. Parker, Jessica S. Horst

**Affiliations:** 1School of Psychological Sciences, University of Newcastle, Callaghan 2308, Australia; 2Research School of Psychology, The Australian National University, Canberra 2601, Australia; jacis7@outlook.com (J.S.); yingxiv@gmail.com (I.Y.J.); emma.vparker@hotmail.com (E.V.P.); 3School of Psychology, University of Sussex, Brighton BN1 9RH, UK; jessica@sussex.ac.uk

**Keywords:** word learning, sleep-related memory, napping, declarative memory, memory strength

## Abstract

Children can easily link a novel word to a novel, unnamed object—something referred to as fast mapping. Despite the ease and speed with which children do this, their memories for novel fast-mapped words can be poor unless they receive memory supports such as further exposure to the words or sleep. Axelsson, Swinton, Winiger, and Horst (2018) found that 2.5-year-old children who napped after fast mapping had better retention of novel words than children who did not nap. Retention declined for those who did not nap. The children received no memory supports and determined the word-object mappings independently. Previous studies report enhanced memories after sleeping in children and adults, but the napping children’s retention in the Axelsson et al. study remained steady across time. We report a follow-up investigation where memory supports are provided after fast mapping to test whether memories would be enhanced following napping. Children’s retention of novel words improved and remained greater than chance; however, there was no nap effect with no significant difference between the children who napped and those who did not. These findings suggest that when memory supports are provided, retention improves, and the word–object mappings remain stable over time. When memory traces are weak and labile, such as after fast mapping, without further memory supports, sleeping soon after helps stabilise and prevent decay of word–object mappings.

## 1. Introduction

Understanding and remembering new words is a key part of language development [1]. Children hear on average 17,000 words per day [2,3,4]. In the first two years of life, their vocabulary expands by up to 300%, and they learn an estimated 10 new words every two weeks [5,6]. When not explicitly told the labels for objects, children use strategies to determine what a speaker is referring to when hearing a novel word [7,8]. When a child hears a novel word (e.g., “Look at the *quokka*”), it could refer to any object, but when in the context of familiar objects (e.g., kangaroos, koalas, wombats), children typically map the word to a novel object (e.g., smiling marsupial). Using the mutual exclusivity assumption [8,9]—where children assume that each object can only have one label—children quickly guess the meaning of a novel word. This is often referred to as “fast mapping” [10] or “referent selection”.

Children are both fast at fast mapping and highly accurate, but when children’s memory for fast-mapped words is tested, performance drops [11,12,13]. Therefore, fast mapping is best described as an initial stage of the word learning process. *Learning* a word involves not only mapping a novel word to its referent but also retaining and recognising the word–object associations after a delay or in different contexts [14]. While this stage is contrastingly slower and more gradual, it can prevent the retention of incorrect associations [15,16]. Retention of fast-mapped words can improve when memory supports are provided after fast mapping, such as repetition of the word with the object or increasing the salience of the object while it is renamed [17,18,19,20]. For example, Horst and Samuelson [13] found increased retention of fast-mapped words in 2-year-old children when each target object was picked up, pointed to and explicitly named after each fast mapping trial. The re-exposure and explicit labelling of the novel objects likely strengthened the word–object associations.

### 1.1. Sleep-Related Memory Consolidation

There is a large body of evidence that sleeps also supports memory, particularly in adults [21,22,23], and more recently, in children and infants [24,25]. Sleep-related memory consolidation is a process where previously encoded information is strengthened and integrated into existing knowledge stores during sleep [26]. Learners typically perform better when tested on previously presented material after a period of sleep compared to the same period of wakefulness [27].

**Active System Consolidation (ASC)**. Sleep is far from a passive state and, as indicated by electroencephalographic measures of brain activity, contains five main stages that are cycled through over 90 min in adults: four stages of non-rapid eye movement (NREM) sleep (Stage 1, Stage 2, Stage 3 slow wave sleep (SWS), Stage 4 SWS), and one rapid eye movement stage (REM) [28,29]. In toddlers, it takes around 75 min to cycle through these stages [30].

The active system consolidation (ASC) theory helps explain why sleep supports memory [31,32,33]. During the initial encoding of information, neural connections are activated in both the neocortex and the hippocampus. These neural connections are initially weak and susceptible to decay [34]. Once SWS begins, these neural connections repeatedly replay in the hippocampus, and this reactivates the neural connections in the neocortex. This also enables the transfer of information from the hippocampus to the relevant neocortical areas, along with an associated decay in the hippocampus. Following transfer to the neocortex, the neural connections are stabilised, and associations with those cortical areas are strengthened [22]. These neural connections are now independent of the hippocampus [33]. SWS supports largely declarative memories, namely memories for explicit information, such as facts, episodes, and semantic information, including the meaning of words [35]. Given the influential role of sleep on memory and toddlers’ difficulties retaining novel fast-mapped words, e.g., [13], the effect of sleep on children’s memory for novel words deserves attention.

### 1.2. Napping and Memory Consolidation

It is not only nocturnal sleep that supports memories but also naps [36]. A nap as short as 6 min can enhance adults’ memories for word lists [37]. Children’s naps are largely made up of SWS and can support young children’s episodic memory [37], visuospatial memory [38], and declarative memory, such as vocabulary [39,40]. In the first two years of life, most children regularly nap, but this declines in the third year [38,41]. As there are large changes in language development in the first few years of life, frequent napping might help increase vocabulary size during this period [42,43].

Evidence that napping promotes word learning is increasing. Horváth, et al. [39] trained 16-month-old infants on two novel word–object pairs. Retention tests two hours later (based on preferential looking to the correct target) revealed that those who napped in the 2-h interval from immediate testing had increased retention, whereas those who remained awake showed no change in retention.

More recently, Williams and Horst [44] presented 3-year-old children with novel words during storybook reading. Children heard the same novel words in either the same story read three times or in three different stories. Following an immediate retention test, half of the children napped before their retention was tested 2.5 h, 24 h, and one week later. Children who napped had significantly higher retention than those who remained awake in both storybook conditions. In addition, children who heard the same stories repeated also outperformed those who heard different stories as word repetition in the same contexts likely aided encoding. Interestingly, retention was similar across time for the children who heard different stories and napped as those who heard the same stories but did not nap. According to Williams and Horst, napping compensated for the more varied presentation of the novel words heard in different stories. The strength of association between the words and objects was perhaps weaker for the varied stories group, and napping after helped strengthen the associations.

Axelsson, et al. [45] tested the effect of napping on 2.5-year-old children’s memory for fast-mapped words. Following an immediate retention test, half of the children napped and half remained awake before a delayed retention test four hours later and another retention test the following morning (see Figure 1). Although immediate retention was the same for both conditions, the children who napped had significantly higher retention than those who did not nap on both delayed retention tests. However, unlike previous studies where performance increased after napping [39,44,46], retention scores remained stable across time in [45]. For the children who did not nap, retention declined significantly by the following morning (see Figure 4B). Therefore, napping helped stabilise memory of fast-mapped words and reduced the rate of decay.

### 1.3. Multiple Factors Affecting Sleep-Related Memory Consolidation

Many factors can affect sleep’s support on memory, such as the quality and duration of sleep, the age of the participants, and the learning material e.g., declarative or non-declarative material involving implicit or procedural memory which is less available to consciousness [27,42]. Other factors include the elapse of time from exposure to the material to the onset of sleep and the strength of encoded memories, both of which will be the focus here.

**Time Interval from Encoding to Sleep**. How soon sleep begins after exposure to learning material could also affect sleep-related memory consolidation. Sleep could have a greater stabilising effect if it occurs soon after encoding [42], particularly for labile associations such as fast-mapped words. One study on infants’ memory for actions [47] and another on memory for word sequences [48] found that nap onsets occurring within 4 h of exposure were associated with significantly greater retention than nap onsets occurring after 4 h. However, there was no clear rationale for the 4-h cut-off. Children were grouped based on whether they slept within 4 h making it difficult to assess the effect of falling asleep shortly after 4 h. In William and Horst’s [44] study on the memory of novel words in storybooks, napping started within 45 min of hearing the stories. The question is whether 45 min is a critical time point for the nap effect they found or whether it would be evident with longer sleep onset intervals. We chose to test children 4 h after exposure to the material as in Hupbach, et al. [48] and Seehagen, et al. [47]. The children in the nap condition in Axelsson, et al. [45] and the current study fell asleep at a variety of times before the first delayed test 4 h later, as did the children in the wake condition for their nocturnal sleep. Children were tested the following morning to test for further delayed retention and when children in both the nap and wake conditions had slept [39]. Time of sleep onset from exposure to learning material is typically studied as a categorical variable rather than a continuous variable. It became apparent during testing for Axelsson, et al. [45] that it would be useful to assess this time as a continuous interval to allow for a more detailed exploration of the variation in sleep onset times. This may help determine optimal times to introduce children to novel material to facilitate memory retention, particularly for weaker associations.

**Strength of Memory Prior to Sleep**. Napping children’s memory for fast-mapped words did not increase in Axelsson, et al. [45]. One possible reason is that the fast-mapped words were weakly encoded. Children’s retention of fast-mapped words is typically poor, and the initial memory traces are too weak to support retention unless children receive memory supports such as ostensive naming after fast mapping [12,13,17]. Axelsson et al.’s [45] findings suggest napping can stabilise and maintain children’s retention for several hours and into the next day. Previous research indicates that the strength of associations prior to sleep can affect the ability to see sleep-related effects [22,42]. For example, Drosopoulos, et al. [49] trained one group of adults with lists of word pairs to 90% accuracy in cued recall. Another group was trained to 60% accuracy. Improvement in cued recall following nocturnal sleep was seen only in the lower accuracy condition, suggesting that the benefits of sleep on memory were only evident for weakly encoded declarative information. Similarly, in a study on procedural motor learning (button press sequences), Wilhelm, et al. [50] found that napping only supported the memories for children (4–6 years) trained to a lower level than a higher level.

Sleep also supports memories subjected to interference, such as when participants are trained on two lists of word pairs; those who sleep shortly after training have better recall for the first list than those who remain awake [49,51]. Sleep likely helps make the original list more resistant to interference. These examples highlight where sleep supports the consolidation of weaker memories. Contrastingly, Tucker and Fishbein [52] split participants into high- and low-scoring groups based on recall of word pairs relative to training performance. A nap effect was only evident in the high-performing group rather than the low-performing group, suggesting naps only support stronger memories. However, it is difficult to compare to other studies where participants were trained to a pre-determined level [49,51], and recall was quantified differently (difference scores from training).

Stickgold [23] further argued that the effect of pre-sleep memory strength follows an inverted-U-shaped curve, with strong and weak memories less supported by sleep, but those encoded to an intermediate level, benefiting most from sleep. There is likely little capacity for neural reactivation during sleep with excessively weakly encoded material, and heavily encoded material would already be integrated into cortical stores. Sleep in both cases would offer little consolidative benefit. However, these studies involved different learning materials, methods, age groups and definitions for memory strength and learning level. This makes it difficult to compare the level of memory strength across the studies.

**Current study**. Here, the effects of sleep onset and the strength of memory for fast-mapped words prior to napping were tested. The fast-mapped words in Axelsson, et al. [45] were possibly weakly encoded as they only fast-mapped and retention only stabilised following napping rather than increased as is typically seen in other studies [44,47,48]. Ostensive naming can increase children’s retention of fast-mapped words [17,20]. What remains unknown is whether ostensive naming would be associated with enhanced retention particularly in those who nap due to stronger pre-nap memory strength [49].

Using the same methods as the Axelsson, et al. [45] study, we exposed 40 toddlers to four novel words and novel objects using a fast mapping task presented on a computer. The only difference to the Axelsson et al. study was that after every referent selection trial, the correct target moved up the screen away from the competitors, and children heard the name repeated (see Figure 2). Horst and Samuelson [13] argued that ostensive naming benefited children’s memory for new words because the target object was distanced from the competitors when it was explicitly named. Retention was tested immediately after fast mapping, after a 4-h interval during which half the children napped, and after nocturnal sleep roughly 24 h later (see Figure 1). We predicted a nap-related memory effect: the children who napped after fast mapping would have better delayed retention than those who remained awake, as in Axelsson, et al. [45]. As the children here saw the objects ostensively named, we also predicted that retention would increase in the children who napped rather than only stabilise. However, if sleep-related memory effects are only found with weaker memories [49,51], then the inclusion of ostensive naming would lead to smaller differences between the nap and wake conditions than what was seen in Axelsson, et al. [45]. To test for the effects of prior memory strength, a comparison to retention in Axelsson et al. [46] was performed as children in that study did not receive ostensive naming. This comparison was expected to reveal better retention in the current study with ostensive naming. When it comes to time intervals from fast mapping, sleep onsets occurred at a variety of times prior to the afternoon test for those who napped and prior to nocturnal sleep for those who did not nap. We aggregated the nap and wake conditions into one group to assess sleep onset intervals as a continuous variable to determine the specific times associated with better retention the following morning. Sleep onsets occurring sooner after novel word exposure was expected to be associated with better post-nocturnal retention, e.g., [48].

## 2. Method

### 2.1. Participants

The final sample had 40 (nap: 20, wake: 20) typically developing, monolingual children (23 girls, 17 boys, *M* age = 30 months, 18 days, see Table 1). To aid in comparability to Axelsson, et al. [45], the same sample size was used. A power analysis for that study using G*Power [53], indicated that a sample size of 38 was needed to detect a difference between the nap and wake condition with a large effect size (0.95), alpha of 0.05 (two-tailed), and power level of 0.80. All children had normal vision. Participants were recruited via social media posts and flyers at childcare centres and libraries. Each participant received a book and gift voucher (10 AUD). A further three children were tested but were excluded due to inattentiveness.

The mothers’ average number of years in education was 16.93 years, and 65% had completed bachelor or post-graduate degrees, 20% a form of higher education, and 15% high school. The fathers’ average number of years in education was 15.57, and 53.8% had completed bachelor or post-graduate degrees, 25.6% a form of higher education, and 20.5% high school. There were no significant differences between the nap and wake conditions in numbers of each gender, age, vocabulary size, nocturnal sleep duration, parents’ age, or parents’ years in education (see Table 1). As retention data (immediate, afternoon, post-nocturnal, see Figures 4 and 5) was compared to the non-ostensive naming group from Axelsson, et al. [45], we checked for any differences between the participants with comparisons on key demographic variables (see Table 1). The children in this ostensive group were 26 days older on average than those in the non-ostensive naming group, and fathers had significantly higher education in the non-ostensive naming group. Habitual napping can affect sleep-related memory e.g., [38,44]. Parents rated children’s nap habits on a scale of 1–5 (1 = always, 2 = usually, 3 = sometimes, 4 = rarely, 5 = never). Children were categorised as habitual if they always or usually napped and non-habitual if they sometimes, rarely, or never napped (see Table 1). Numbers of habitual nappers were identical across the ostensive and non-ostensive groups.

### 2.2. Apparatus and Materials

The objects in the fast mapping and immediate retention trials were presented on a 24-inch Dell monitor (1920 × 1080 resolution) using Experiment Builder (1.10.1630; SR Research) and an EyeLink 1000 eye-tracking system. Eye-tracking data was collected for a separate study.

To determine the sleep durations and sleep onset times, CamNtech Actiwatches (MotionWatch 8) were used along with parent-recorded sleep diaries. The actigraphy watches estimate periods of sleep/wake based on bodily movement with an accelerometer (Ancoli-Israel et al., 2003). To calculate the time interval from fast mapping to sleep onset, the time stamp of the eye-tracking recording was subtracted from the actigraphy variable “Fell asleep”. To measure productive vocabulary size, parents completed the Australian English Communicative Development Inventory OZI [54]. The test-retest reliability of the OZI is high, along with measures of internal reliability. The Maldonado Pictorial Sleepiness Scale [55] was used to measure sleepiness at the delayed retention tests. Children are presented with five cartoon faces with increasing depictions of sleepiness (1–5). This was used to determine if poor performance could be explained by sleepiness. Even though it is aimed at young children, it is uncertain children understand the scale, so we also asked parents to rate their child’s sleepiness and used the parents’ ratings as done in [45].

**Stimuli**. The familiar objects used for familiarisation and fast mapping (*n* = 58) were sourced from Shutterstock.com and the Bank of Standardised Stimuli [56]. The choice of familiar objects was based on words typically known by 2.5-year-olds according to the Australian English Developmental Inventory [54]. Four novel words and images were sourced from the Novel Object and Unusual Name (NOUN) Database [57]. Four novel objects were presented because in previous studies without ostensive naming, 30-month-old children’s immediate retention of 4 novel objects (0.45–0.60) was greater than chance (0.25) with a medium effect size [45,58]. This was also to aid in comparability to Axelsson, et al. [45]. All images were on average of 53 × 79 mm (5.05 × 7.53° at a 60 cm distance).

### 2.3. Design and Procedure

Ethical approval was obtained from the Human Research Ethics Committee at The Australian National University. The method and design were identical to Axelsson, et al. [45] except that we included ostensive naming after each fast mapping trial. Children attended three testing sessions: the first at the university where fast mapping and immediate retention were completed, and two delayed retention tests at the child’s home. The first delayed retention test was completed around four hours (*M* = 4.25 h, *SD* = 0.82) after immediate retention (“afternoon”), and the second test was the following morning (“post-nocturnal”; *M* = 21.91 h after fast mapping, *SD* = 1.32).

**Warm-up**. To ensure children comprehended the task requirements before commencing, they were asked to point to pictures of familiar objects (e.g., bee, elephant, ladder) on three A3 posters on a wall. Experimenters praised correct choices and provided guidance when needed.

**Familiarisation trials**. The same type of training was repeated except now in front of a computer monitor and eye tracker. Children sat either on a caregiver’s lap or in a car seat or booster chair 60 cm away from the monitor and 55 cm from the eye-tracking camera. In each trial, children saw three familiar objects, and an audio recording asked children, “*Can you see the__? Point to the__. Where is the__?*” This repeated until the child pointed or for a maximum of 30 s. The experimenter ended the trial once the child made the selection. To ensure children were centrally fixated before each trial, an animated attention-getter appeared in the centre of the screen, and trials only commenced once children looked at it for 300 ms. Experimenters praised correct choices and provided guidance when needed. If unsuccessful, children were given a break before re-attempting.

**Fast mapping**. The same procedure was used in the fast mapping trials but with no feedback. Two familiar objects and one novel object appeared in each trial (see Figure 2). The same audio instructions as in the familiarisation trials were used *(“Can you see the__? Point to the__. Where is the__?”*). There were eight familiar target trials, which alternated with eight novel target trials. Familiar target trials were included to ensure that children were selecting objects based on the object labels. Across the eight novel target trials, the four novel objects were the target object twice but never in succession. Familiar competitors changed in every trial. After each fast mapping trial, the target object moved up the screen and bounced four times, and the object’s name was repeated, “*Can you see the noop?*” There were four presentation orders of the experiment, counterbalancing the appearance order of each novel object (first, second, third, fourth) and their positions across trials (left, middle, right).

**Immediate retention**. As four novel words were tested, the layout changed with an object in each quadrant of the screen (see Figure 3). To familiarise children with the new layout, four familiar objects were presented, and participants were asked to find a familiar target. This was followed by four novel target retention trials with only the novel targets appearing on the screen. The presentation order of the novel targets was the same as during fast mapping. The position of the novel objects changed in each trial. The auditory instructions were the same as those used during fast mapping, and no feedback was provided. Before leaving the university, each child was fitted with the actigraphy watch and parents were provided with sleep diaries.

**Afternoon and post-nocturnal retention**. Delayed retention was tested in the child’s home approximately 4 h later. Within the interval from immediate testing to afternoon, children either napped or remained awake. We assigned children to the nap or wake condition quasi-randomly based on whether parents expected their child to nap or not in the following 4 h. However, as children’s tendency to nap decreases from around 24 months of age [41], we allowed napping to occur on the children’s own accord. All the children assigned to the nap condition napped, and two children assigned to the wake condition napped and were moved to the nap condition. Testing continued until there were 20 in the nap and 20 in the wake condition. The reason for allowing the children to do this was to prevent depriving regular nappers of a nap or imposing napping on those who no longer nap. Further, if children need a nap and are prevented from doing so, this could enhance poor performance due to fatigue [38,59]. If only habitual nappers are assigned to the nap condition, sleep-related effects might be amplified as napping habits can enhance sleep-related memory [38,44]. Finally, in some studies, children were presented with material immediately before their typical nap time [38,44,48]. With the current group, as the children could nap at any time in the following four-hour interval, the variability of sleep onset times was enhanced, allowing for a test of the effect of time interval (from exposure to fast-mapped words to sleep onset) on delayed retention.

Using an iPad (programmed using Xcode 6), the experimenter presented four familiarisation and four novel retention trials. Familiarisation trials involved four trials with familiar targets to train children to select targets using the iPad. These were repeated if necessary. Novel retention trials involved four trials with novel targets, appearing in the same order and layout as in the immediate retention test. The only difference between the immediate retention and afternoon retention tests was that the audio recording asked children to touch rather than point to the target object (e.g., “*Can you see the noop? Touch the noop. Where is the noop?*”). Children received no feedback. Post-nocturnal retention occurred the following morning after nocturnal sleep, using the same procedure.

## 3. Results

First, planned comparisons were performed, using jamovi (v2), comparing retention to chance level performance at each retention test, using one-sample *t*-tests. Planned comparisons between the nap and wake conditions were also performed using independent *t*-tests. To test for any changes in retention across testing sessions for the nap and wake conditions as well as between the groups in the ostensive and non-ostensive naming studies, linear mixed-effects analyses were performed with the GAMLj module 2.0.1 [60] in jamovi. This module was developed in R [61] with R’s lme4 package [62]. Maximum likelihood was used to estimate the parameters as we were interested in the fixed effects. The default Type III ANOVA-style *F*-tests were used, and degrees of freedom were calculated using the Satterthwaite method. The effect of each variable was assessed within models along with comparisons between models using likelihood ratio tests (LRT) based on the models’ –2 log-likelihood (−2LL) values (presented in Appendix A). The default convergence optimiser (bobyqa) was used.

**Ostensive Naming: Nap and Wake Planned Comparisons**. Fast mapping was first compared to chance in the nap and wake conditions (see Appendix A in Appendix A). All were significantly different from chance levels except the nap condition at immediate testing (see Figure 4A). There were no significant differences in retention between the nap and wake conditions at the immediate test (*t* (38) = −1.11, *p* = 0.275, *d* = 0.35), afternoon test (*t* (38) = −0.57, *p* = 0.575, *d* = 0.18), and post-nocturnal test (*t* (38) = −0.84, *p* = 0.409, *d* = 0.26). Comparisons using Welch’s *t* revealed that the children in the wake condition were significantly sleepier at the afternoon retention test than the children in the nap condition (see Table 1). The difference in sleepiness was non-significant at the post-nocturnal test.

### 3.1. Retention across Immediate, Afternoon, and Post-Nocturnal Tests

**Ostensive Naming Group.** Using stepwise forward selection, each variable was added to the linear mixed models (see Appendix A Appendix A for the full list of models). There was no overall main effect of condition (nap, wake), but when test session (immediate, afternoon, post-nocturnal) was entered with polynomial contrasts, there was a significant quadratic trend (see Appendix A in Appendix A for the parameter estimates) indicating that retention increased from immediate to afternoon testing, which levelled off overnight to post-nocturnal testing (see Figure 5). The interaction between condition and test session was non-significant, suggesting that the change in retention scores between the nap and wake conditions did not differ significantly (see Appendix A).

**Comparison Between Ostensive and Non-Ostensive Naming Studies.** To determine whether hearing the objects explicitly labelled or not after fast mapping affected retention, the variable group (ostensive naming, non-ostensive naming) and the interactions between groups by condition (nap, wake), groups by test session, and groups by test session by condition were added to the above models (see Appendix A Appendix A for the full set of models). All were significant aside from the final interaction (see Appendix A in Appendix A for parameter estimates). Follow-up simple effects tests, revealed that the condition by group interaction was explained by larger overall retention scores in the wake condition in the ostensive naming (*M* = 0.51, *SE* = 0.04) than the non-ostensive naming group (*M* = 0.29, *SE* = 0.04), *t* (239.67) = −4.33, *p* < 0.001. The difference between the nap conditions in the ostensive naming (*M* = 0.45, *SE* = 0.04) and the non-ostensive naming group (*M* = 0.46, *SE* = 0.04) was non-significant, *t* (236.50) = 0.09, *p* = 0.932. The significant group by test session interaction was based on different directions in polynomial trends across the two studies. In the ostensive naming group, there was an overall increase in retention from immediate to afternoon testing and stabilisation to post-nocturnal testing, but there was a reduction in retention in the non-ostensive naming group (see Figure 5).

### 3.2. Post-Nocturnal Retention: Time Interval from Encoding to Sleep Onset

The children in these studies fell asleep at a variety of times in the four hours prior to afternoon testing (nap condition) and beyond for those who did not nap (see Table 1 for the mean time intervals and Appendix A in Appendix A). Therefore, it is unclear whether falling asleep closer to encoding plays an important role in delayed retention. Despite no clear difference in retention between the children who napped or not in the ostensive naming group, cluster analyses were performed based on sleep onset time from encoding rather than grouping based on nap occurrence. Therefore, the key variable here was the duration of time from exposure to the novel words to when they first fell asleep. Rather than assigning the participants based on whether they napped or not, this analysis aimed to determine whether participants fell into groups based on the degree of similarity in the time interval from fast mapping to sleep onset and subsequent post-nocturnal retention. This will also help to determine the sleep onset intervals associated with post-nocturnal retention in the ostensive and non-ostensive naming studies.

**Post-nocturnal Retention: Ostensive Naming Group**. A hierarchical cluster analysis was performed using IBM SPSS 27 to determine the number of clusters the participants fell into based on the time interval from encoding to sleep onset (in minutes) and post-nocturnal retention. Scores were standardised (*z*-scores) due to the differing scales (minutes and accuracy proportion). Euclidean distance was used to measure the similarities between the scores, and Ward’s method was used to create the clusters. A dendrograms was used to judge the number of clusters (see Appendix A Appendix A). This indicated that there were two main clusters.

Based on the two clusters suggested by the hierarchical cluster analysis, a K-means cluster analysis was performed, which grouped participants based on the time from fast mapping to nap onset (time interval) and post-nocturnal retention and compared the groups (see Table 2). Children who fell asleep on average 2 h and 56 min after encoding had an average post-nocturnal retention score of 0.46. In contrast, children who fell asleep on average 9 h and 23 min after encoding had an average post-nocturnal retention score of 0.65. Thus, falling asleep later was associated with better post-nocturnal retention.

**Post-nocturnal Retention: Non-ostensive Group**. The hierarchical cluster analysis revealed that there were two main clusters (see Appendix A in Appendix A). The follow-up K-means cluster analysis indicated that those with lower post-nocturnal retention (0.19) fell asleep on average 8 h 40 min after encoding, whereas those with higher post-nocturnal retention (0.38) fell asleep around 1 h and 50 min after encoding. Therefore, when there is no ostensive naming, sooner sleep onset times are associated with better retention.

## 4. Discussion

We tested the effect of napping on children’s delayed memory for novel words. Our methods were identical to Axelsson, et al. [45] except that children heard the novel words ostensively named after each fast mapping trial. Axelsson, et al. [45] found significantly greater retention for non-ostensively named words at the afternoon and post-nocturnal tests in the children who napped after fast mapping than those who remained awake. In contrast, in the ostensive naming group of the current study, we did not find differences in retention between the nap and wake conditions. For both conditions, retention increased from immediate to afternoon testing and levelled off overnight. Children’s retention was also significantly greater than chance for both conditions at afternoon and post-nocturnal retention. Including ostensive naming in the fast mapping task was expected to enhance nap-related memory due to an increase in memory strength of novel word–object associations. Nap-related memory enhancement is typically found in other studies, e.g., [39,44]. Here, both children who napped and remained awake benefitted from ostensive naming.

### 4.1. Prior Exposure before Sleep

The only methodological difference between the current study and Axelsson, et al. [45] was the inclusion of ostensive naming. Without ostensive naming, the memory traces were likely weaker in Axelsson, et al. [45]. In that study retention in the wake condition declined over time, but retention in the nap condition remained steady, suggesting that napping soon after exposure helped stabilise rather than enhance retention. With ostensive naming, word–object associations may be stronger at encoding, and even though there was no effect of napping (better performance in the nap vs. wake condition), retention increased and remained steady overnight. The lack of a nap-related memory effect (better performance in nap condition) cannot be explained by sleepiness, as it was the children in the wake condition who were significantly sleepier than children in the nap condition at the afternoon test. The difference in sleepiness was non-significant at post-nocturnal testing.

In both studies, for the children who napped, retention of the novel words at the delayed retention tests was significantly greater than chance. For the children who stayed awake, delayed retention was significantly greater than chance among the children who received ostensive naming only. Without any support, such as napping or ostensive naming, retention fares poorly.

In previous studies with adults, memory strength or prior knowledge was manipulated by varying the amount of training or initial encoding prior to sleep. Some studies find that an effect of sleep is only found for those trained to or at lower levels of accuracy prior to sleep [49,51]. The findings here suggest that when it comes to fast mapping in toddlers, a nap-related effect is only evident with less exposure to the novel words, such as when children only fast map and no further labelling of the objects is provided. Greater exposure to the word–object associations with fast mapping plus ostensive naming likely leads to stronger memory traces that are more robust to forgetting in later retention tests [17,63]. However, a nap-related effect (better performance in the nap vs. wake condition) was not evident. The quadratic trend seen with the inclusion of ostensive naming suggests that nocturnal sleep is also important in maintaining children’s retention as both the nap and wake conditions were above chance the following morning.

### 4.2. Time Interval from Exposure to Sleep Onset

The timing of sleep onset following exposure to new material is also important for memory retention as some studies have found that nap-related memory consolidation is only found when young children sleep within four hours of exposure to new material [38,44,47,48]. The rationale for the four-hour time point is unclear. Is there much difference in post-nocturnal retention when children fall asleep slightly before or after the four-hour cut-off? We investigated the interval of time from fast mapping to sleep onset as a continuous variable. As the children were free to sleep at any point, this allowed for a range of times from fast mapping and until nocturnal sleep times. Rather than group children based on whether they napped or not within four hours from fast mapping, we instead aggregated the nap and wake conditions and performed a hierarchical cluster analysis based on sleep onset interval time and post-nocturnal retention scores. For the ostensive naming group, falling asleep on average 2 h and 56 min after fast mapping was associated with lower average post-nocturnal retention when compared to those falling asleep on average 9 h and 23 min later. For the study where there was no ostensive naming, children who fell asleep on average around 1 h and 50 min after encoding had higher average post-nocturnal retention than those falling asleep later at around 8 h 40 min. The question then is whether falling asleep around 2 h is critical for memory consolidation of fast-mapped words when no other memory supports are provided. Certainly, more research is required to determine if this is the case, but timing is likely to be important with sooner sleep onsets, particularly for weaker associations that could decay when no there is no repetition or clear labelling of novel objects. This is similar to findings with adults [64] and for children with and without language comprehension difficulties [65]. The results with better retention following longer sleep onset intervals in the ostensive naming group were unexpected, but it could have been because there was no nap effect. The associations were likely stronger in this group, and the time interval of sleep onset was possibly less critical for delayed retention. Henderson, et al. [66] found that children’s memory for novel words heard in storybooks was better if the stories were heard 3–5 h before sleep rather than just before sleep, suggesting that there could be ideal time intervals with certain types of material. With the addition of ostensive naming after each fast-mapping trial, children not only heard the correct word–object associations after each fast-mapping trial, this also meant they received greater exposure time seeing and hearing the word–object associations than the non-ostensive group. This likely made the word–object associations sufficiently strong that wake-related memory consolidation occurred, and later sleep onsets were more beneficial as has been seen during the restful wake in adults [67,68,69]. Riggins and Spencer [70] found an association between hippocampal volume and memory; those who were no longer habitually napping had better performance after a longer period of wakefulness. In the ostensive group, 55% of the children with later sleep onsets (those in the wake condition) were no longer habitually napping. Interactions between memory strengths and sleep onset intervals are worthy of further investigation, along with the type of learning material and nap habituality.

### 4.3. Implications for Word Learning

#### Fast Mapping and Ostensive Naming

Word learning is a memory process involving the transformation of newly formed associations into associations stored in long-term memory that can be retrieved later [16,63]. When children are exposed to new words in ambiguous situations, they typically use prior knowledge such as existing vocabulary to eliminate familiar words and select the most novel objects as the most likely referents of novel words [18,71]. Given children’s poorer delayed memory of fast-mapped words [11,12,13], fast mapping is best described as an initial step of a more gradual learning process [11,15]. Given that additional memory supports help novel word retention without sleep [13,20], and sleep can support children’s memory, investigating the relationship between fast mapping, ostensive naming, and napping can contribute to our understanding of children’s vocabulary acquisition. The findings here suggest that when children only engage in fast mapping, the opportunity to nap and ideally within 2 h of exposure the fast-mapped words, word–object associations are maintained. When there is no opportunity for repeated exposure or clear labelling, associations are more likely to decay. Caregivers are not always able to label all objects for children, so napping could help to support vocabulary acquisition for words that children have merely guessed or had little exposure to.

When clear object labelling is provided, napping is less critical, but later nocturnal sleep can help to maintain those word–object associations. When considering both studies, napping appeared to stabilise retention in similar ways. The difference lay more in those who did not nap—those who did not hear the objects labelled after fast mapping had a greater level of memory decay than those who heard the words labelled. The high proportion of SWS and sleep spindles (small bursts of activity occurring in quick succession) during naps can explain why napping supports children’s retention of weaker memories such as novel words that have been heard via guessing [38,72]. Wakefulness and nocturnal sleep later appear to support stronger associations (see also effects of restful wake in adults [67,68,69].

These findings could have implications for children with developmental disorders, particularly if children experience problems with attention and/or sleep, see [73] for a review. For example, sleep-disordered breathing not only disrupts sleep but can also reduce SWS and declarative memory consolidation [74]. As children’s naps contain high levels of SWS and SWS supports declarative memory [38,75], providing children with sleep difficulties the opportunity to nap could support declarative learning. Clear labelling and repetition of new words could help with long-term retention, particularly if children suffer from attention difficulties and if regular napping is less likely.

### 4.4. Limitations

One possible limitation to the current study is that we assumed that the addition of ostensive naming strengthened children’s word–object associations [13,17]. The strength of learned associations was not controlled for explicitly as in studies with adults by training participants to pre-determined levels [49]. This could be controlled for, even with toddlers, by varying accuracy proportion levels required of participants before assigning them to conditions. This could also aid in determining optimal strengths of word–object associations that benefit from sleep-related memory consolidation. Future research should explore methods to control for learned association strength in toddlers.

Immediate retention performance did not differ significantly between the ostensive and non-ostensive studies, but they did at the delayed retention tests. Therefore, the effects of ostensive naming became apparent later. Regardless, Soderstrom and Bjork [76] point out the need to distinguish between learning and performance—performance is not always a useful indicator of learning. Participants can perform poorly despite substantial learning, and participants can demonstrate strong performance despite poor learning. Orban, et al. [77] found memory-related changes in brain activity using fMRI despite no clear changes in performance. What does this mean for the fast mapping studies investigated here? Although children’s immediate retention in the ostensive naming group was not much better than those in the non-ostensive naming group, the superior memory effect was yet to happen. Schönauer, et al. [78] argued that during a period of wakefulness, the hippocampus provides a form of temporary protection for memory, and this seems to be the case for the ostensively named fast-mapped words in the wake condition but not for those who did not receive ostensive naming. This might also explain the unexpected time interval effect in the ostensive group. When memories are stronger, they remain in temporary hippocampal store until sleep-onset and this possibly also explains a lack of a nap effect (better performance in nap vs. wake condition) in the ostensive naming group. Alternatively, a lack of nap effect could have been due to sufficient exposure in the ostensive naming group, which might have led to integration into neocortical lexical networks that are less prone to interference [27]. Nonetheless, nocturnal sleep likely supported memory consolidation in the ostensive naming group due to the stabilisation in retention from afternoon to post-nocturnal sleep in both the nap and wake conditions. This did not occur in the non-ostensive naming group in the wake condition. Further studies with the addition of measurements of neural activity will be beneficial for understanding children’s acquisition of the new words and unpacking the relationship between performance and retention of fast-mapped words.

Something to also consider is the number of items presented. Four novel objects were presented as in Axelsson et al. [46] but also because 30-month-old children’s immediate retention was moderate without ostensive naming. However, with ostensive naming, both the nap and wake condition had greater than chance delayed retention. The difficulty level was possibly too low, and a greater number of novel items could provide greater power to discriminate performance between the nap and wake conditions. Studies with eight novel items revealed that with ostensive naming, 24-month-old children retain around four [13]. This would be a suitable number for future studies.

Another limitation is that the two studies do not provide a direct comparison of the effect of fast mapping and ostensive naming as the ostensive naming group experienced both fast mapping and ostensive naming. A clearer comparison between the effects of the two could be found in a study where the novel words are only ostensively named. This might also lead to more equal levels of word–object associations between fast mapping and ostensive naming. Zosh, et al. [79] found greater than chance retention of fast-mapped words but chance level retention of explicitly named words and argued that the inferences involved in fast mapping and the processing of competitors aid in word learning. Contrastingly, Himmer, et al. [80] found a stronger sleep-related effect (better memory after sleep) in explicitly named words than fast-mapped words in adults, arguing that fast-mapped words were already well integrated into long-term memory prior to sleep but see [81]. Clearer comparisons of the effects of fast mapping and explicit naming in relation to sleep-related memory are needed.

Other limitations to consider are that the children were assigned to the nap and wake conditions based on whether parents expected their child to nap and whether they napped or not before the afternoon test. However, there were unequal numbers of habitual nappers in each condition (nap = 17; wake = 11). The children in the wake condition were also sleepier in the afternoon than those in the nap condition, yet there was no significant difference in retention between the nap and wake conditions. Kurdziel, et al. [38] found that habitual nappers had greater nap-related memory than non-habitual nappers. Sandoval, et al. [82] found no difference in nap-related memory between habitual and non-habitual nappers and concluded that this might be because their task (generalisation of verbs) was more difficult than Kurdziel et al.’s task (picture-location memory). One question is whether the lack of a nap effect here is because there was a greater number of non-habitual nappers in the wake condition (nap = 3; wake = 9). Lam, et al. [83] found that napping was associated with poorer nocturnal sleep and lower vocabulary size. Other studies have found that less consolidated sleep (i.e., still napping) was associated with poorer vocabulary size later in development [84,85] and better delayed memory [70]. The greater number of non-habitual nappers in the wake condition raises the question of whether they were more cognitively advanced than the children in the nap condition that had more habitual nappers. The participants here had similar vocabulary sizes (see Table 1), which can affect novel word retention [27], but it is uncertain if children in the wake and nap conditions in the ostensive naming group differed in some way that may have masked any nap-related memory effects. In future studies, equal numbers of habitual and non-habitual nappers should be randomly assigned to the nap and wake conditions, see [82]. Finally, the classification of nap habituality here was based on descriptive terms (i.e., always to never). Other studies define habituality as napping 4–5 days or more per week [38,40,70,82], and this might be easier for parents to determine and aid in comparability to other studies.

There was also a significant age difference (26 days) and difference in fathers’ education (1.5 years) between the ostensive and non-ostensive groups, but the difference in vocabulary size and the number of habitual nappers between the groups was non-significant. The relationships between age and retention at the three test points (*r_s_* (38) = 0.02–0.17, *p_s_* = 0.124–0.859), and between fathers’ education and retention were non-significant (*r_s_* (38) = −0.16–0.04, *p_s_* = 0.184–0.725). Nonetheless, future studies should also include cognitive measures or standardised tests of development to ensure that children in the nap and wake conditions have equal potential to acquire new words.

## 5. Conclusions

Children are often faced with new information, and it is unlikely that sleep strengthens all newly formed memory traces equally as this could overwhelm memory systems [86]. As sleep involves an active consolidation process [33], after word–object associations are encoded and stored temporarily in the hippocampus, certain memory representations might be selectively reactivated during periods of SWS and transferred to long-term memory stores. A nap effect (better retention in children who nap than those who do not) is more evident for weaker initial word–object associations, and earlier sleep onsets following exposure are needed to maintain word–object associations and prevent decay. With stronger word–object associations, as a result of clear labelling following fast mapping, wake-related reactivations can occur, resulting in better-delayed retention, and weakening of word–object associations is less likely. Later sleep onset intervals can be more suitable for stronger word–object associations particularly, see also [66] when napping regularity starts to decline [70]. Multiple factors affect sleep-related memory [42], and the findings here suggest that when it comes to fast-mapped words, the strength of encoding and the time intervals from exposure to sleep onset play a role in delayed retention.

## Figures and Tables

**Figure 1 brainsci-11-01366-f001:**
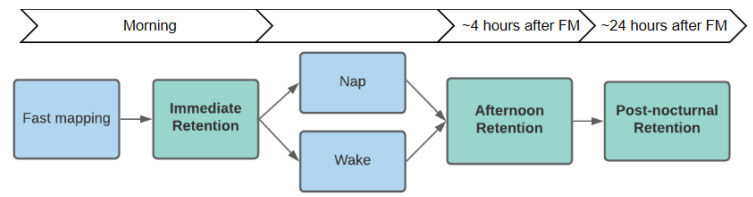
Time Course from Fast Mapping and the Ensuing Retention Phases.

**Figure 2 brainsci-11-01366-f002:**
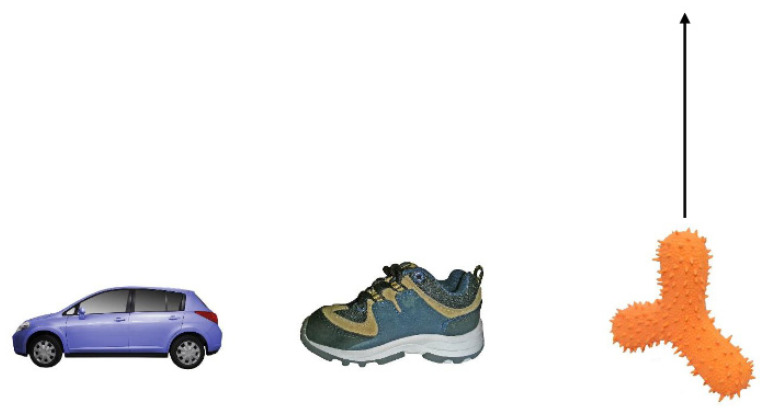
Example Fast Mapping Trial with Two Familiar Objects and One Novel Object. Arrow Indicates the Destination of Object During Ostensive Naming.

**Figure 3 brainsci-11-01366-f003:**
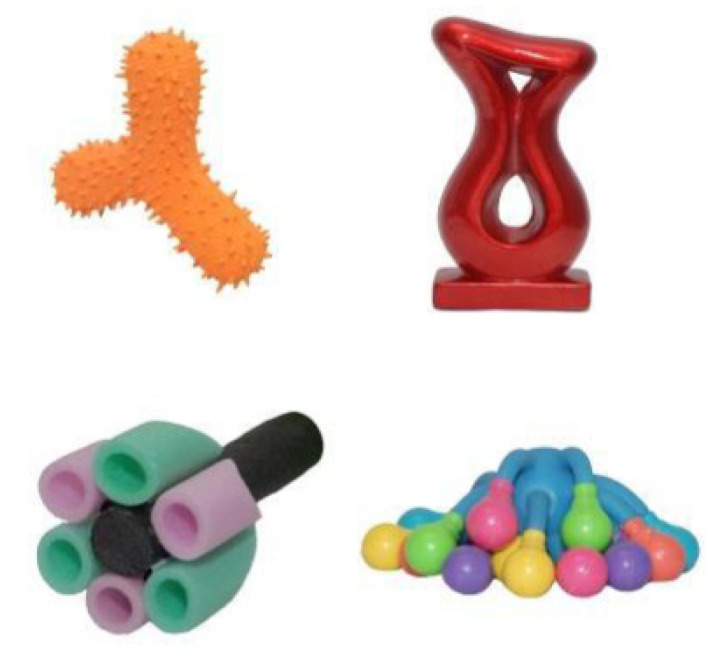
Display Layout in the Retention Tests for the Four Novel Objects.

**Figure 4 brainsci-11-01366-f004:**
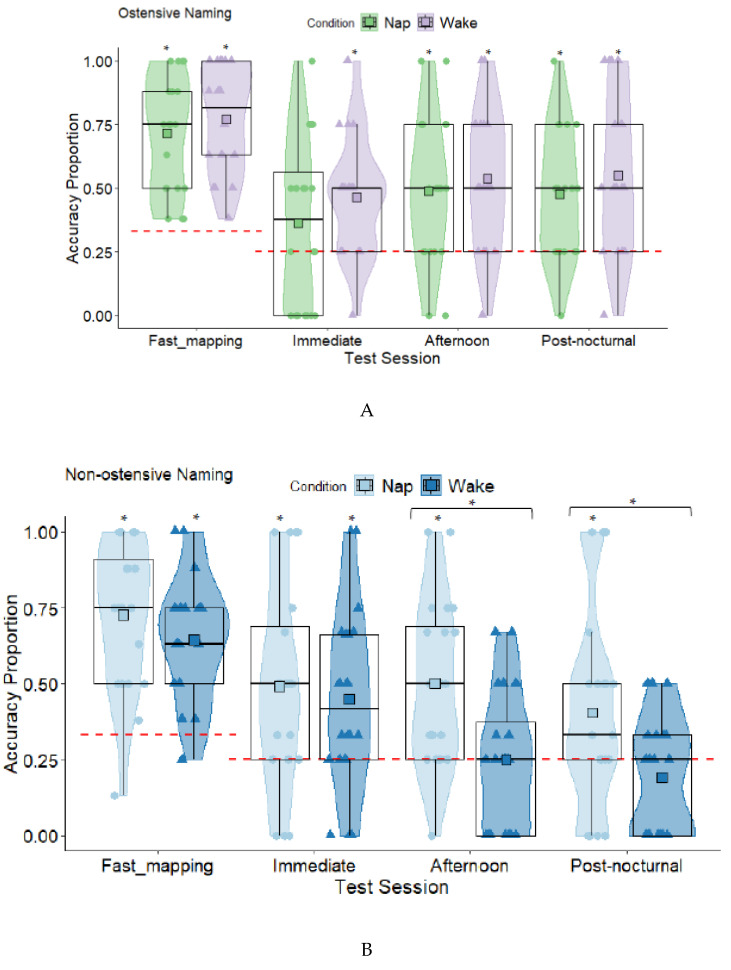
Box and Violin Plots for Accuracy During Fast Mapping, and Immediate, Afternoon, and Post-nocturnal Retention Tests for the Ostensive (**A**) and Non-ostensive Naming Groups ((**B**) from Axelsson et al. [46] presented here for comparison)(Squares Denote Means; Red Dashed Lines Denote Chance Level, * = *p* < 0.05).

**Figure 5 brainsci-11-01366-f005:**
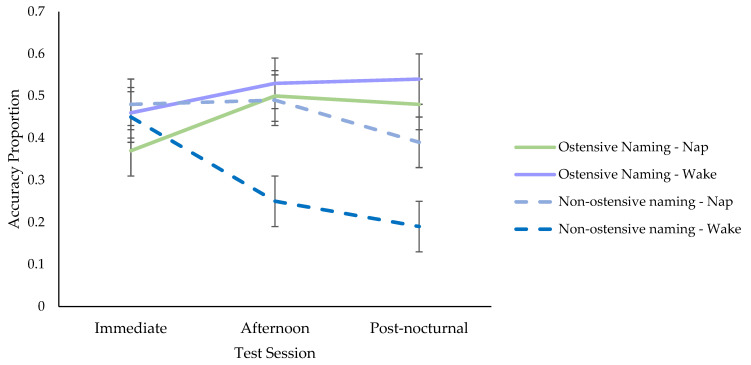
Retention Across Test Sessions in the Ostensive Naming and Non-Ostensive Naming Groups.

**Table 1 brainsci-11-01366-t001:** Sociodemographic Variable Comparisons.

	Ostensive Naming Group					Group Comparison				
	Nap Condition		Wake Condition			Ostensive Naming		Non-Ostensive Naming (Axelsson et al., 2018)		
Participant										
Gender (% *F*)	60		55			58		50		
Habitual napper (%)	85		55			70		70		
Age range	28 m 10 d–32 m 28 d		28 m 3 d–32 m 27 d			28 m 3 d–32 m 28 d		28 m 8 d–31 m 27 d		
	*M*	*SD*	*M*	*SD*	*t*	*M*	*SD*	*M*	*SD*	*t*
Age	30 m, 18 d	1 m, 8 d	30 m, 18 d	1 m, 11 d	0.29	30 m, 18 d	1 m, 13 d	29 m, 22 d	30 d	−3.14 **
Vocab. (OZI)	291.25	60.79	286.05	74.82	0.24	289	67	302	48	−1.01
Nap duration	1 h, 11 m	42 m	NA	NA	NA	1 h, 11 m	42 m	1 h, 22 m	30 m	−0.91
Nocturnal Sleep	8 h, 13 m	1 h, 14 m	8 h, 48 m	1 h, 18 m	−1.32	8 h, 13 m	1 h 14 m	7 h, 59 m	1 h, 2 m	0.65
Sleepiness										
Afternoon	1.30	0.47	2.95	1.00	−6.68 ***	2.13	1.14	2.64	1.06	−2.08 *
Post-nocturnal	1.20	0.52	1.45	0.51	-1.53	1.32	0.53	1.68	0.97	−2.00
Time Interval from Fast Mapping to First Sleep Onset										
Hours	2.42	1.32	7.33	2.84	−7.02 ***	4.87	3.31	4.75	3.60	0.16
Mother										
Age (years)	36.53	5.68	34.23	4.72	1.40	35.38	5.28	33.86	3.65	1.50
Education (years)	17.79	2.66	16.08	3.03	1.85	16.93	2.94	17.79	2.01	−1.50
Father										
Age (years)	38.30	9.29	35.55	4.84	1.17	36.89	7.39	35.48	3.77	1.07
Education (*years*)	15.83	3.09	15.32	3.30	0.49	15.57	3.17	17.04	2.47	2.25 *

* *p* < 0.05; ** *p* < 0.01; *** *p* < 0.001.

**Table 2 brainsci-11-01366-t002:** K-means Cluster Analyses for the Ostensive and Non-Ostensive Naming Studies.

	Cluster 1	Cluster 2	*F*	*df*	*p*
Ostensive Naming Time Interval (mins) (Hours) Post-nocturnal retention *n*	563 (9.38) 0.65 12	176 (2.93) 0.46 28	172.38 4.12	38 38	<0.001 0.050
Non-ostensive Naming Time Interval (mins) (Hours) Post-nocturnal retention *n*	521 (8.68) 0.19 17	111 (1.85) 0.38 23	352.29 5.08	38 38	<0.001 0.030

## Data Availability

Publicly available datasets were analysed in this study. This data can be found here: https://osf.io/b824k/ (accessed on 12 October 2021).

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
