# Peer review of "Prior Exposure and Toddlers’ Sleep-Related Memory for Novel Words"

_brainsci, 2021, doi:10.3390/brainsci11101366_

Round 1

Reviewer 1 Report

I was interested to read this paper, which is on a very worthy topic with broad appeal and potential for theoretical advance. However, interpretation of the results is complicated by under-powered analyses and key confounds (i.e., the nap versus wake effects are confounded by a weighting of habitual nappers in the nap group versus the wake group; the timing effects are confounded by the additional sleep that nap participants receive prior to the post nocturnal test). Introduction The introduction is largely well written and well informed. I only have a few suggestions. In the abstract you state “ These findings suggest that when memory supports are provided, retention improves, and sleep helps to maintain word-object mappings over time.” I think the latter part of this interpretation is an overstatement, as you don’t know here whether sleep is maintaining word-object mappings from the afternoon to the post nocturnal test, or whether simple passing of time would be as effective. “As there are large changes in language development in the first few years of life, frequent napping might help increase vocabulary size during this period (Diekelmann, Wilhelm, & Born, 100 2009; Horváth & Plunkett, 2016).” It would be useful to also acknowledge literature that suggests that the consolidation of daytime to nighttime sleep predicts language development over the preschool years. (e.g., https://academic.oup.com/sleep/article/34/8/987/2454717; https://doi.org/10.1017/S0305000920000677) These studies are relevant to the section on timing: https://doi.org/10.1111/jcpp.13253; https://www.sciencedirect.com/science/article/pii/S0022096521001259. They suggest that, in older children, learning closer to sleep (but not at bedtime) may be beneficial (with the first study suggesting that this might be particularly the case for children with weaker language abilities). These findings may be more relevant in the discussion. “Stickgold (2009) further argued that the effect of pre-sleep memory strength follows an inverted-U shaped curve, with strong and weak memories less supported by sleep, but those encoded to an intermediate level, benefiting most from sleep.” It would be helpful to provide more neuroscientific rationale for the inverted-U shaped curve. The key hypotheses are difficult to navigate. On the one hand you’re predicting an increase in retention following sleep, owing to the ostensive naming procedure (that wasn’t used in the previous Alexsson et al study where memory retained rather than improved over sleep), BUT contrastingly, you’re also predicting that because the memories will be stronger the sleep-benefits will be reduced. It’s not clear to me whether these are conflicting hypotheses or not. For instance, ostensive naming might improve retention in both wake and sleep conditions, which might be partly responsible for the reduced sleep-benefit? This sentence is also unclear “A comparison to retention in Axelsson et 212 al. (2018) was expected to reveal better retention in the current study with ostensive naming.” - it’s not clear at this point whether you are going to perform a cross-experiment analysis, or not. There’s very little information on the timing hypothesis, and without further information on the methods at this point (i.e., the timings that were compared) it’s difficult to comprehend the following hypothesis: “When it comes to timing, sleep onsets occurring sooner after exposure were expected to be associated with better post-nocturnal retention (e.g., Hupbach et al., 2009).” - in the introduction to the “Current study” it would be helpful to say how timing was manipulated, so that this makes more sense, and by “sleep onsets” do you mean nap onsets or nocturnal sleep onsets? Methods What was the age range of the sample? How was vocabulary size measured (as listed in Table 1)? Line 229: “As retention was compared to the non-ostensive naming group [FROM] Axelsson et al. (2018),”. Can you also say a little bit more about the data that was taken from Axelsson et al (e.g., was all data available used?). I understand why you didn’t force toddlers into a nap/wake condition and instead allowed them to nap naturally (or not). However, this is somewhat concerning, not only because more habitual nappers ended up in the nap group (owing to them being more likely to go to sleep), but also because we know that the consolidation of naps to nighttime sleep at this age is a predictor of language ability at school entry, and thus, the nap group may have less mature language development (this is perhaps somewhat supported by the below change performance at immediate test for the nap group? Even though there’s no group difference, this still seems relevant to consider). Perhaps the lack of difference in the vocabulary size measure (reported in Table 1) could help to perturb this concern, although if it was a parent report vocabulary measure then it might not be so sensitive. Also, in your additional analysis, you find that performance benefits were found for toddlers who stayed awake / napped in line with their habitual behaviour, and thus, this means that the majority of the nap group (85%) were conforming to their typical behaviour, whilst only 45% of the wake group were conforming to their typical behaviour (given that 55% were habitual nappers). A larger sample size with balance of habitual versus non-habitual behaviours in each group could have offered a stronger means of controlling and understanding this potential confound. Line 325: it would be useful to provide descriptive stats on the time that elapsed between learning and nap onset here (and perhaps also learning and nocturnal sleep onset), to give us an indication of how much variability there was. I would have found a procedure figure helpful in the Methods, making clearer when each of the tests were administered. I also think you need to include information on the timing (means/SDs) of the immediate and delayed tests for each group. It’s stated that the post nap test was administered roughly 4 hours after the immediate test, but it would be helpful to know whether the groups were matched in this regard. The same information would be helpful for the post-nocturnal sleep test - were the groups matched on the timing of this test? Results/Discussion The added notation to Figure 4 (signalling significant contrasts) looks to have become misaligned with the bars. Line 380: Please can you clarify the ns for the analysis that looks at habitual versus non-habitual nappers (perhaps in Figure 5). Given there were only 20 per nap/wake group, we are working with very small ns here, which severely questions the extent to which this analysis is worthy of inclusion / meaningful. Also, the error bars on Figure 5 (which need labelling) look to be very much overlapping for the immediate and post nap tests for the non-habitual nappers, and so the “upward quadratic trend” for the wake group looks to be non-existent. Generally then, I’m struggling to know how to interpret the habitual/nonhabitual analysis, and where this analysis leaves us with interpreting the main results. The significant group by test interaction in the ostensive/non-ostensive naming analysis is complicated by the “post-nap” condition label, when this test did not occur “post-nap” for half of the sample. Can Figure 6 (and the description in the text) be clarified to reflect this? E.G., you could use “post nap/wake” on the Figure instead of “Post nap”, or even “afternoon test” as in Axelsson et al. The timing analysis is really interesting but to me could suggest that those who are habitually staying awake are showing better post-nocturnal retention (if they are ostensive naming at least, and initial traces are strong enough to allow for wake-based reactivation prior to nocturnal sleep, perhaps). This is also in line with evidence that toddlers who consolidate their daytime naps sooner have better language development. Whereas when learning opportunities are more sparse (i.e., no ostensive naming) then sleeping soon after learning is more important (this aligns with the James et al., 2021 reference I mentioned above) Line 531-4 “These results were unexpected, but could be because there was no nap effect in the ostensive naming group.” This needs more explanation, it’s difficult to unpack. It should be clearer from the outset that the contrast between the ostensive naming and no-ostensive naming datasets isn’t just about the inclusion of this procedure, it could also be about the level of exposure (given the two conditions are presumably not matched on the number of exposures to each item? A number of important limitations are carefully mentioned, but it will also be important to acknowledge the small sample sizes (which are not pre-justified with a power analysis) and the very small number of items, both of which will contribute to low statistical power (particularly for the additional analyses). On the whole, the study feels very exploratory rather than confirmatory (i.e., hypotheses are not strongly theoretically justified, particularly for the timing analysis - for which it feels as though the study wasn’t really set up to address - and have not been pre-registered) which should be acknowledged. Clearer statements of the theoretical contributions that the paper can make would help to strengthen the discussion. Typos Line 190: typo “Here, were test…” Line 208: “As the children here saw the words ostensively named, we also predicted that retention increase in the children who napped rather than only stabilised.” needs to be “..retention would increase….. rather than only stabilize”. Discussion: Under “Prior exposure before sleep” the first two paragraphs feel quite repetitive. Line 561 “engage is fast mapping” typo here.

Reviewer 2 Report

This study tested the effects of memory supports, in the form of ostensive labeling, on retention of words learned through fast-mapping as compared to results of a prior study that did not provide memory supports. Both studies considered the effects of sleep vs wake on retention in 2.5 year olds at immediate test, 4 hours post nap or post nocturnal sleep. Napping resulted in increased retention and a reduced rate of decay over time when no memory supports were provided (re-presentation of a prior published result). When memory supports are provided, retention was greater than in the prior study with equal retention in sleep and wake groups, contributing new findings to the literature on sleep and word learning. I have a few comments.

Is there a justification for why the experimenters test children at 4- and 24-hr intervals after learning? If so, it would be helpful to convey this.

Regarding sleep benefits as a function of encoding level, it is difficult to compare Tucker and Fishbein (2008) who found a greater effect for subjects with higher levels of encoding with studies that show that lower levels of encoding benefit more from a nap. Although Tucker and Fishbein used a median split to identify groups of subjects with greater and lesser amounts of retention after encoding, they reported difference scores in their papers, not encoding levels, making it difficult to understand what “high encoding” means relative to the other studies that trained subjects to specific levels of encoding performance. It is important to make this point so as to not confuse the results in the literature. As it is impossible to know how high vs low encoding levels in Tucker and Fishbein map onto other papers in the literature, the authors could de-emphasize or remove the study from further discussion in the remainder of the paper.

Children in the ostensive naming group were significantly older than children in the non-ostensive naming group. How might this affect outcomes?

Fathers in the non-ostensive naming group had significantly higher education than those in the ostensive naming group. How might this affect outcomes?

The experimenters rated nap habituality in a different way than other studies in the literature that ask how many naps children take in a week. For instance, what does it mean for different parents that children nap usually, sometimes or never? It is important to point this out as well as the fact that the authors used the same assessment in the original non-ostensive naming study.

A limitation of this study that should be addressed is that children were not randomly assigned to nap and wake groups making it difficult to know whether another variable may account for the experimental outcomes. The authors point out some benefits of not assigning children to nap/wake groups and include variables that might interact with the outcomes, but the authors should still acknowledge this limitation.

One last issue to address has to do with nap habituality within the cluster analyses. Riggins and Spencer recently showed relationships between hippocampal volume and performance in a memory task as a function of nap habituality that might predict that children who perform well after a long period of wake might be nonhabitual nappers.

Round 2

Reviewer 1 Report

The authors have down a sound job of attending to the reviews. This paper will make an exciting and valuable contribution to the literature. I strongly recommend acceptance.